# Visualizing drug binding interactions using microcrystal electron diffraction

Max T. B. Clabbers [1], S. Zoë Fisher [2,3], Mathieu Coinçon[4,5], Xiaodong Zou [1] & Hongyi Xu [1✉]

Visualizing ligand binding interactions is important for structure-based drug design and fragment-based screening methods. Rapid and uniform soaking with potentially reduced lattice defects make small macromolecular crystals attractive targets for studying drug binding using microcrystal electron diffraction (MicroED). However, so far no drug binding interactions could unambiguously be resolved by electron diffraction alone. Here, we use MicroED to study the binding of a sulfonamide inhibitor to human carbonic anhydrase iso-form II (HCA II). We show that MicroED data can efficiently be collected on a conventional transmission electron microscope from thin hydrated microcrystals soaked with the clinical drug acetazolamide (AZM). The data are of high enough quality to unequivocally fit and resolve the bound inhibitor. We anticipate MicroED can play an important role in facilitating in-house fragment screening for drug discovery, complementing existing methods in structural biology such as X-ray and neutron diffraction.

[1] Department of Materials and Environmental Chemistry, Stockholm University, 106 91 Stockholm, Sweden. [2] European Spallation Source ERIC, 224 84 Lund, Sweden. [3] Department of Biology, Lund University, 223 62 Lund, Sweden. [4] Department of Biochemistry and Biophysics, 106 91 Stockholm, Sweden. [5] SciLifeLab, 171 65 Solna, Sweden. ✉email: hongyi.xu@mmk.su.se

Small three-dimensional crystals are highly suitable for structure determination by electron diffraction, commonly referred to as microcrystal electron diffraction (MicroED)[1,2] and 3D electron diffraction (3D ED)[3]. MicroED data can efficiently be collected in-house on a conventional transmission electron microscope (TEM) using the rotation method[4,5] from thin hydrated macromolecular crystals. Furthermore, small macromolecular crystals may have less defects and lower mosaicity than larger crystals, and any external changes such as ligand soaking or rapid flash-cooling can be applied faster and more uniformly[6,7]. Small crystal volumes also have their disadvantages, notably the overall diffracting intensity is much weaker, and radiation damage is limiting the maximum attainable resolution and affecting data and model quality[8]. In recent years, MicroED has emerged as a promising method for structural biology, determining structures of several known macromolecules[1,2,5,9–13], and even solving a previously unknown protein structure using MicroED data[14]. These results illustrate how MicroED can complement existing methods in structural biology, loosening the size limitations imposed on the sample as (sub-)micron-sized 3D crystals resist structure determination by X-ray and especially neutron diffraction[15]. Furthermore, biomolecules of low molecular weight that are still challenging for single-particle cryo-EM[15,16] can be studied by MicroED.

An important application of structural biology is structure-based drug discovery and design, as it relies on detailed structural knowledge of the protein-active site and the underlying molecular interactions of small-molecule binding[17]. Similarly, high-throughput screening of a large number of fragments for possible interactions with the target protein can reveal binding interactions that inform the design of novel inhibitors[18–21]. The potential of MicroED for drug discovery was first indicated in recent work studying the binding of the inhibitor bevirimat (BVM) to the C-terminal domain of the HIV Gag protein[22]. The structural model provided insight into the underlying interactions in inhibitor binding. A unique binding pose could however not be resolved from the MicroED data alone. This was further complicated by the fact that the binding site is located on a symmetry axis at the center of the homo-multimer[22]. We previously presented the structure of a novel R2lox ligand-binding metalloenzyme[14]. Although the ligand could not be resolved from the MicroED data, the substrate-binding pocket was reshaped and had an altered projected electrostatic contact potential distribution, indicating a different ligand-binding interaction compared with structural homologs.

Building upon earlier results, we use MicroED to investigate drug-binding interactions to the active site of human carbonic anhydrase isoform II (HCA II), a small ubiquitous metalloenzyme with a molecular weight of 29 kDa. Carbonic anhydrases catalyze the reversible reaction of $CO_2$ hydration to produce $HCO_3^-$ and $H^+$. The reaction has two components: $CO_2$ hydration and the rate-limiting proton transfer step. After the first half-reaction of $CO_2$ hydration, a zinc-bound water molecule (ZW) is left bound to the active-site metal (Zn). In the next step, the ZW is deprotonated to $OH^-$, ready for a nucleophilic attack on the next incoming $CO_2$ molecule. The generated proton ($H^+$) of the ZW deprotonation is transported to the bulk solvent via a hydrogen-bonded water network and an internal proton-shuttling residue, His64[23,24]. There are 15 expressed human carbonic anhydrase isoforms (HCAs) showing some diversity in physiological and subcellular distribution. Of these, HCA II is found in red blood cells and has the highest activity of any HCAs with a $k_{cat}$ of $10^{-6}\,s^{-1}$ and a $k_{cat}/K_M$ approaching the diffusion limit ($10^8\,M^{-1}\,s^{-1}$)[25]. The structural model of the native water-bound HCA II has been well characterized in the past, both by X-ray and neutron diffraction[26,27].

There are many known small-molecular inhibitors of carbonic anhydrases, from small anions to the widely prescribed sulfonamide-based inhibitors used to treat glaucoma, altitude sickness, and congestive heart failure[23,28]. Several crystal structures of carbonic anhydrase complexes with sulfonamide inhibitors show binding interactions that can effectively shut down the enzyme: the ionized –NH group binds directly to the Zn, the –NH group donates a hydrogen bond to the hydroxyl group of Thr199 side chain, the sulfonamide O interacts with the amide backbone of Thr199, and finally binding of the inhibitor displaces the ZW[29,30]. Neutron studies of HCA II in complex with various sulfonamides also revealed additional hydrogen-bonding interactions and water displacements in the active site that are important determinants in understanding inhibitor binding[30–32]. Systemically, inhibition of extra- and intracellular carbonic anhydrases in the kidney causes increased excretion of bicarbonate, other ions ($Na^+$, $K^+$, and $Cl^-$), and water through the urine. This leads to a mild diuretic effect and general metabolic acidosis with associated alkalization of urine[33]. For the treatment of glaucoma, inhibitors are applied as topical eyedrops. Here inhibitors block carbonic anhydrases in the ciliary epithelial cells, disrupting ionic transport across cell membranes, leading to reduced bicarbonate, $Na^+$, and water transport across epithelium cells into the eye. The net effect is that less aqueous humor is formed, reducing intralocular pressure[34].

Here, we present the structure of HCA II in complex with the inhibitor acetazolamide (AZM) demonstrating that MicroED data are of sufficient quality to fit and resolve ligand binding. We demonstrate that drug binding can be studied efficiently by soaking the 3D microcrystals with inhibitor and collecting MicroED data using a conventional TEM, making it effectively feasible to screen for possible protein-inhibitor-binding interactions on a home source. Data processing, structure determination, and fitting and refining the inhibitor bound to the active site was feasible using standard crystallographic routines. We validate the structural model by comparison with the native structure as a negative control to confirm the correct interpretation for the model of HCA II with the bound inhibitor. Furthermore, we compare our MicroED model and data with previously determined X-ray and neutron crystal structures of the same HCA II: AZM complex.

## Results

**Data collection**. Inhibitor-bound protein complexes were obtained after 20 min of soaking HCA II microcrystals with the inhibitor AZM. MicroED data were collected of both the native and inhibitor-bound HCA II crystals using continuous rotation. Microcrystals selected for data collection typically did not exceed a thickness of 500 nm, and diffracted beyond 2.5 Å resolution (Fig. 1). In an attempt to optimize the signal-to-noise ratio and improve the resolution, we collected data over small angular ranges of on average 10–30° with a relatively high dose rate of $0.1\,e^-/Å^2/s$. The accumulated electron dose was typically within $2.0–6.0\,e^-/Å^2$ per dataset in order to minimize radiation damage[8,35].

**Data processing**. Data were integrated and merged to 2.5 Å resolution for structure determination, covering a combined angular range of −60 to +60° (Table 1, see Supplementary Tables 1 and 2). The weighted average of the unit-cell parameters is close to the values described in the literature[29,30]. Owing to the preferred crystal orientation, and limited tilt range of the TEM goniometer stage, overall data completeness did not increase beyond about 73% and 80% for the native and ligand-bound protein, respectively (Table 1). The data are anisotropic

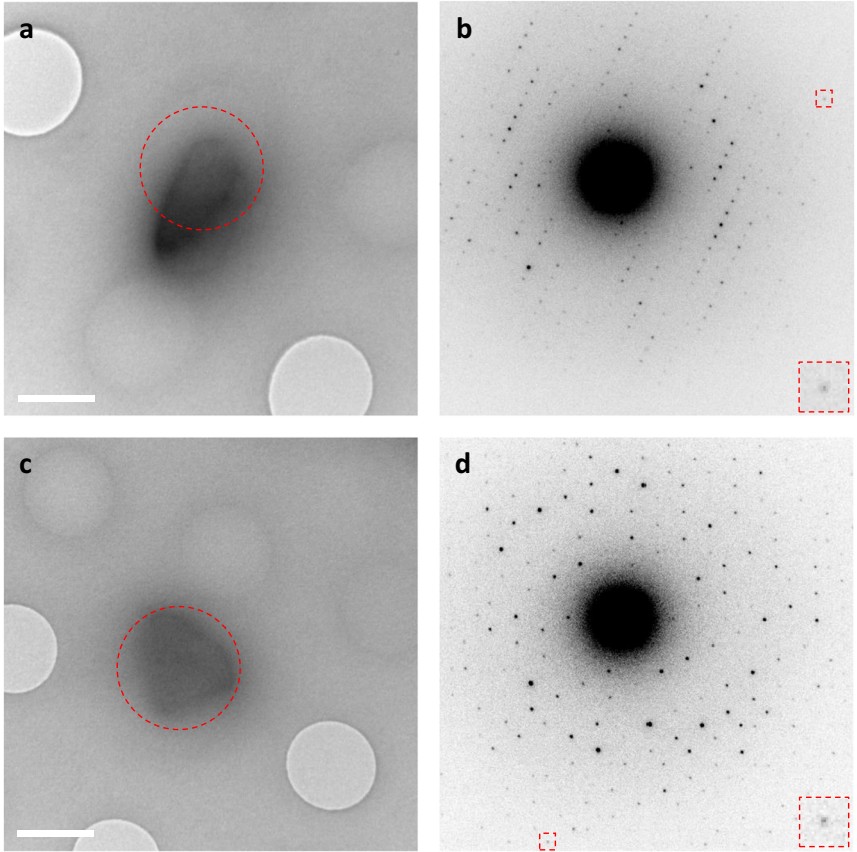

**Fig. 1 Hydrated 3D microcrystals in a thin layer of vitrified ice. a** Micrograph of a native HCA II microcrystal taken at +20.0° tilt (scale bar is 1 μm). **b** Electron-diffraction pattern of native HCA II recorded over 0.68° tilt at a dose of 0.15 e⁻/Å². The inset shows a Bragg spot at 2.5 Å resolution. **c** Micrograph of an inhibitor-bound HCA II:AZM complex microcrystal taken at 0.0° tilt (scale bar is 1 μm). **d** Electron-diffraction pattern of HCA II:AZM recorded over 0.68° tilt at a dose of 0.15 e⁻/Å². The inset shows a Bragg spot at 2.3 Å resolution. Red circles indicate the area of 1.5 μm in diameter selected for MicroED data collection (selected-area electron-diffraction mode).

predominantly along the direction of $c^*$ because of orientation bias. Completeness decreases in the lowest- and highest-resolution bins owing to shading of the reflections by the direct beam and varying crystallinity and diffracted intensity for different crystals, respectively (see Supplementary Tables 3 and 4).

**Structure solution and refinement**. The structures of native HCA II and HCA II:AZM complex were phased by molecular replacement, using an apo structure of HCA II with all ligands and solvent removed as search model[29]. A well-contrasting single solution was found in space group $P2_1$ using maximum-likelihood molecular replacement in Phaser[36].

Following rigid-body refinement of the molecular replacement solution, the resulting electrostatic potential map for the native structure shows clear difference potential, indicating the presence of the $Zn^{2+}$ metal cofactor (Fig. 2a). The structure was inspected and modeled using Coot[37], fitting the zinc metal cofactor coordinated to the three active-site histidine residues (His94, His96, and His119), and refined against the MicroED data (Fig. 2b). The final structural model has a $R_{work}/R_{free}$ of 0.249/0.276 (Table 1), and a coordinate precision (maximum-likelihood estimate) of 0.41 Å.

For the inhibitor-bound complex, clear difference potential can be observed for the active-site zinc and the AZM inhibitor at its expected position and orientation where it displaces the solvent and zinc-bound water (Fig. 2c). The calculated mFo–DFc difference potential map was then used to fit the AZM inhibitor using Coot[37] at a contour level of 2.8σ (Fig. 2d). A final structural

model was obtained after restrained reciprocal space refinement of the initial inhibitor fit, fixing several geometry outliers and placing solvent molecules. The model has a final $R_{work}/R_{free}$ of 0.224/0.255 (Table 1), and a reported coordinate precision (maximum-likelihood estimate) of 0.37 Å. To further assess the coordinate precision of our model, we refined the structure ten times using each time a different random seed with shaking of the atomic coordinates by random perturbations of 0.3 Å. This resulted in a core root-mean-square (r.m.s.) deviation of the atomic coordinates for the parallel refinements of 0.052 Å for the protein backbone (257 residues), and a r.m.s. deviation of 0.073 Å for the inhibitor (14 atoms).

We confirmed that the observed difference potential map used for fitting the inhibitor is not an artifact of model bias by comparing the structure of native HCA II with the structure of HCA II:AZM. The difference potential map of the native structure is only indicative of the metal cofactor, and no significant difference signal is observed for the location where the inhibitor is expected to bind (Fig. 2).

**Inhibitor-binding interactions**. The drug inhibitor AZM has a high affinity for binding HCA II with a $K_i$ of 10 nM[38]. Furthermore, the active site of HCA II is well accessible for binding of potential inhibitors that block the catalytic activity of the enzyme. The inhibitor-binding interactions of AZM are primarily facilitated by its sulfonamide group that replaces the zinc-bound water in the active site. The MicroED structure of inhibitor-bound HCA II does indeed show an active-site zinc that is tetrahedrally

**Table 1 Data collection and refinement statistics.**

| | HCA II[a] | HCA II:AZM[b] |
|---|---|---|
| *Data collection* | | |
| Space group | $P2_1$ | $P2_1$ |
| Cell dimensions[c] | | |
| $a, b, c$ (Å) | 42.51(1), 41.30(1), 72.79(2) | 42.55(2), 41.53(1), 72.11(4) |
| $\alpha, \beta, \gamma$ (°) | 90.00, 104.58 (3), 90.00 | 90.00, 104.62 (3), 90.00 |
| Resolution (Å) | 29.14–2.50 (2.56–2.50) | 35.69–2.50 (2.57–2.50) |
| $R_{merge}$ | 0.285 (0.945) | 0.272 (0.877) |
| $I/\sigma I$ | 4.8 (1.2) | 4.7 (1.5) |
| $CC_{1/2}$ | 0.976 (0.465) | 0.959 (0.584) |
| Completeness (%) | 72.6 (60.0) | 80.0 (71.1) |
| Redundancy | 5.9 (4.1) | 4.4 (2.8) |
| *Refinement* | | |
| Resolution (Å) | 29.14–2.50 | 35.69–2.50 |
| No. of reflections | 6292 | 6895 |
| $R_{work}/R_{free}$ | 0.249/0.276 | 0.224/0.255 |
| No. of atoms | | |
| Protein | 2049 | 2049 |
| Ligand/ion | 1 | 18 |
| Water | 11 | 27 |
| *B* factor (Å$^2$) | | |
| Protein | 27.07 | 20.65 |
| Ligand/ion | 24.07 | 20.58 |
| Water | 26.73 | 20.59 |
| R.m.s. deviations | | |
| Bond lengths (Å) | 0.002 | 0.002 |
| Bond angles (°) | 0.496 | 0.683 |
| Ramachandran | | |
| Favored (%) | 96.08 | 96.08 |
| Allowed (%) | 3.92 | 3.92 |
| Outliers (%) | 0.00 | 0.00 |
| Clashscore | 4.20 | 3.68 |
| Rotamer outliers (%) | 0.00 | 0.00 |

Values in parentheses are for the highest-resolution shell. Data were truncated at approximately $I/\sigma I \geq 1.0$ and $CC_{1/2} \geq 0.4$ with a correlation significant at the 0.1% level[48].
[a]Merged data from 12 crystals (see Supplementary Tables 1 and 3).
[b]Merged data from 13 crystals (see Supplementary Tables 2 and 4).
[c]Values in parenthesis show estimated standard deviation for unit-cell parameters.

coordinated with the –NH group of the sulfonamide inhibitor and three histidine residues (Fig. 3a). The histidine-to-zinc distance is 2.0 Å, which is consistent with previous observations of 2.0 and 2.2 Å from X-ray (PDB ID 3hs4)[29] and neutron diffraction (PDB ID 4g0c)[30], respectively. The lone pair of the sulfonamide N is coordinated directly to the metal cofactor, at a distance of 2.1 Å. This is comparable to available X-ray and neutron models with distances of 1.9 and 2.4 Å, respectively. In addition, the hydrogen of the sulfonamide N can act as hydrogen donor to Thr199 (nitrogen-to-oxygen distance of 2.6 Å) that in turn acts as a donor to Glu106 (oxygen-to-oxygen distance of 2.6 Å). The *B* factors for the ligand are within the same range as those of the more stable side chains of the active-site residues (Table 1).

**Structure validation**. The refined structural model of the HCA II: AZM shows that all features of the active site are well resolved from the electrostatic potential map (Fig. 3b). To validate the structural model with the bound inhibitor, a simulated annealing (SA) composite omit map was calculated, covering the entire unit cell. The SA composite omit map agrees well with the interpretation of the inhibitor-bound protein model (Fig. 3c).

**Comparing MicroED with X-ray and neutron-diffraction data**. We compare our model and electrostatic potential map against previously solved structures of the same complex from joint refinement of neutron and X-ray diffraction data (PDB ID 4g0c[30]) and X-ray diffraction data alone (PDB ID 3hs4[29]) (Fig. 4). The coordination of the metal cofactor and inhibitor in the active site in our model is highly similar to the X-ray and neutron models. Although the MicroED data have lower resolution (2.5 Å) and completeness (80%) used in refinement compared with those of the X-ray diffraction data (1.1 Å and 95%), our electrostatic potential map can still show all important features, such as well-resolved side-chain potential and the coordination of the inhibitor bound to the active site (Fig. 4). This is similar to what could be expected from an electron-density map using X-ray diffraction data at similar resolution and completeness. The nuclear-density map from joint refinement at slightly higher resolution (2.0 Å) and completeness (86%) is less well resolved than our electrostatic potential map, but does provide information about the position of hydrogen atoms in the active site that are difficult to resolve from X-ray data alone. To compare the binding poses of the ligand, we aligned the MicroED, neutron, and X-ray models with respect to the zinc atom and its coordination sphere (Fig. 4d). We calculated the r.m.s. deviation in atomic coordinates of the AZM ligand (14 atoms) aligned with respect to the zinc atom. We found a r.m.s. deviation of 0.466 Å for the X-ray model with respect to the MicroED structure, and 0.679 Å for neutron model. Between the X-ray and neutron model, the r.m.s. deviation in atomic coordinates is 0.409 Å.

## Discussion
We show that MicroED data can effectively be used for visualizing protein-inhibitor-binding interactions by determining the structure of HCA II in complex with clinical drug AZM (Fig. 2). The electrostatic potential map was of sufficient quality to allow accurate model building to resolve ligand binding. We confirmed the correct interpretation for modeling the structure with the bound inhibitor by analyzing the distances for binding interactions, and evaluating our model and data against previously determined structures of the same complex (Figs. 3 and 4). Since the structure was phased by molecular replacement, and as data completeness is limited, the structure may be biased by the search model. We present SA composite omit maps that agree well with the HCA II:AZM model, showing well-defined electrostatic potential for the main chain, side chains, and inhibitor. No missing reflections were filled in for map calculations from weighted estimates of calculated structure factors.

The results presented here demonstrate that MicroED has the potential to resolve inhibitor binding, and may play a significant role in future drug-discovery experiments. At 2.5 Å resolution, our MicroED data seem suitable for fragment-based screening to identify potential protein inhibitors[21]. Furthermore, MicroED data can efficiently be collected from hydrated 3D microcrystals using the rotation method on a conventional TEM, enabling in-house diffraction experiments on a home source for screening and structure determination. MicroED can thereby complement existing methods in macromolecular X-ray and neutron crystallography when size requirements dictate the use of larger crystals. In addition, working with protein microcrystals may have several advantages, such as less sample material required and fast diffusion of ligand. We anticipate that with future hardware and software development optimized toward automated and high-throughput data collection and processing[35,39,40], MicroED can become even more competitive with the ease and speed of synchrotron data collection and fragment screening.

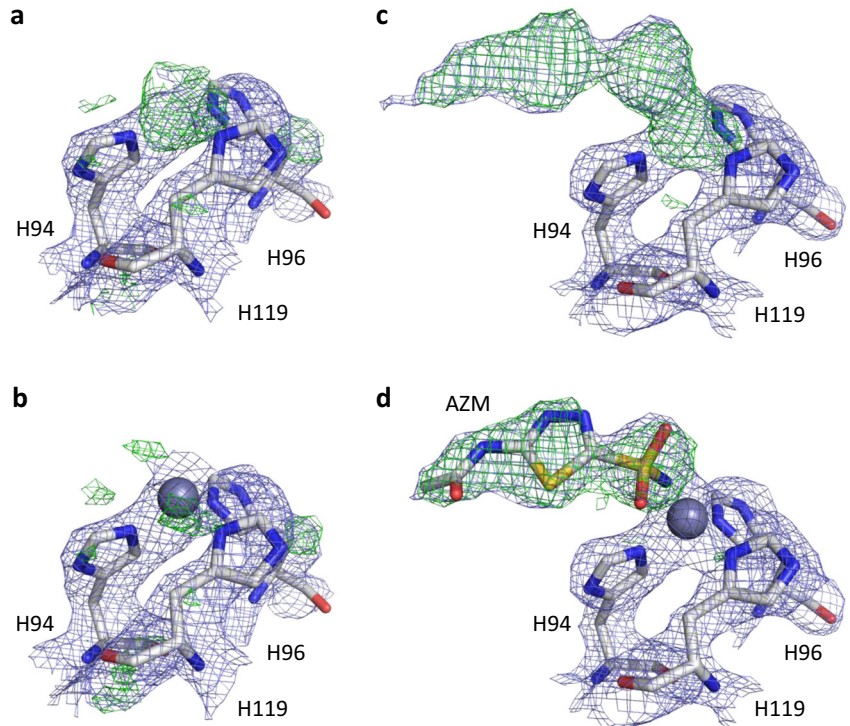

**Fig. 2 Structure solution and ligand fitting.** Electrostatic potential maps displayed for residues of the HCA II active site for the native structure (**a**, **b**) and the AZM inhibitor-bound model (**c**, **d**). **a** Initial map after rigid-body refinement of the molecular replacement solution for the native model, showing clear difference potential indicative of the $Zn^{2+}$ metal cofactor. **b** Refined native model after main-chain rebuilding and fitting the metal cofactor in the difference map. **c** Initial map after rigid-body refinement of the molecular replacement solution for the ligand-bound model, showing clear difference potential indicative of the $Zn^{2+}$ metal cofactor and AZM inhibitor. **d** The difference map after rebuilding of the main chain and fitting of the metal was used for placing the drug inhibitor; the found and fitted ligand is superimposed on the structure model, showing its fit to the mFo–DFc map. Electrostatic potential maps 2mFo–DFc are contoured at $1.2\sigma$, colored in blue, and difference maps mFo–DFc contoured at $2.8\sigma$, colored in green and red for positive and negative peaks, respectively. Only observed data were used, and no missing reflections were restored for map calculations. Carbon, nitrogen, oxygen, and sulfur atoms are colored in gray, blue, red, and yellow, respectively. Zinc is shown as a dark-gray sphere.

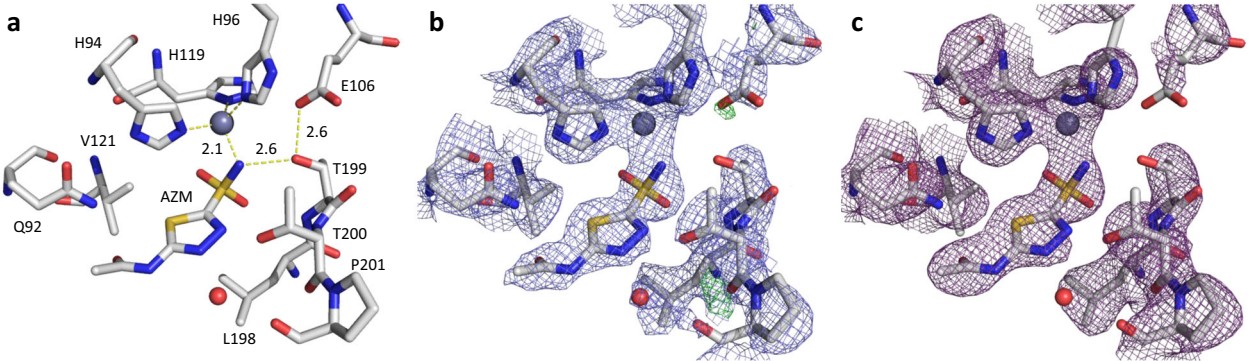

**Fig. 3 Binding interactions of the AZM inhibitor to the active site of HCA II. a** Interatomic distances measured for binding of the sulfonamide group of AZM, the lone pair of the sulfonamide nitrogen is at a distance of 2.1 Å to the active-site zinc, and hydrogen bonding to Thr199 is facilitated at 2.6 Å distance measured from nitrogen to oxygen. **b** Electrostatic potential map 2mFo–DFc contoured at $1.2\sigma$, colored in blue, and difference potential map mFo–DFc contoured at $2.8\sigma$, colored in green and red for positive and negative peaks, respectively. **c** Simulated annealing (SA) composite omit map calculated with sequentially omitting 5% fractions of the structure; the SA composite omit map is contoured at $1.2\sigma$, colored in magenta. For panels **a**–**c**, carbon, nitrogen, oxygen, and sulfur atoms are colored in gray, blue, red, and yellow, respectively. Zinc is shown as a dark-gray sphere, water shown as a red sphere. Only observed data were used, and no missing reflections were restored for map calculations.

We compare our MicroED data against electron- and nuclear-density maps obtained by X-ray and neutron diffraction, respectively (Fig. 4). Our electrostatic potential map is well-resolved despite limited resolution and low data completeness. Electrons offer certain advantages, for example, the possibility to refine atomic charges, exploring the charged state of protein–ligand-binding interactions[10,41]. Furthermore, electrons have improved contrast for visualizing hydrogen atoms compared with X-ray diffraction, and observed hydrogen-bond lengths are closer to the more accurate internuclei distances for neutron scattering[42,43]. With additional improvements in instrumentation and data collection strategies, resolution and data completeness can be improved, and increase the accuracy and precision of electron-diffraction data[35,44]. Given these prospects, MicroED

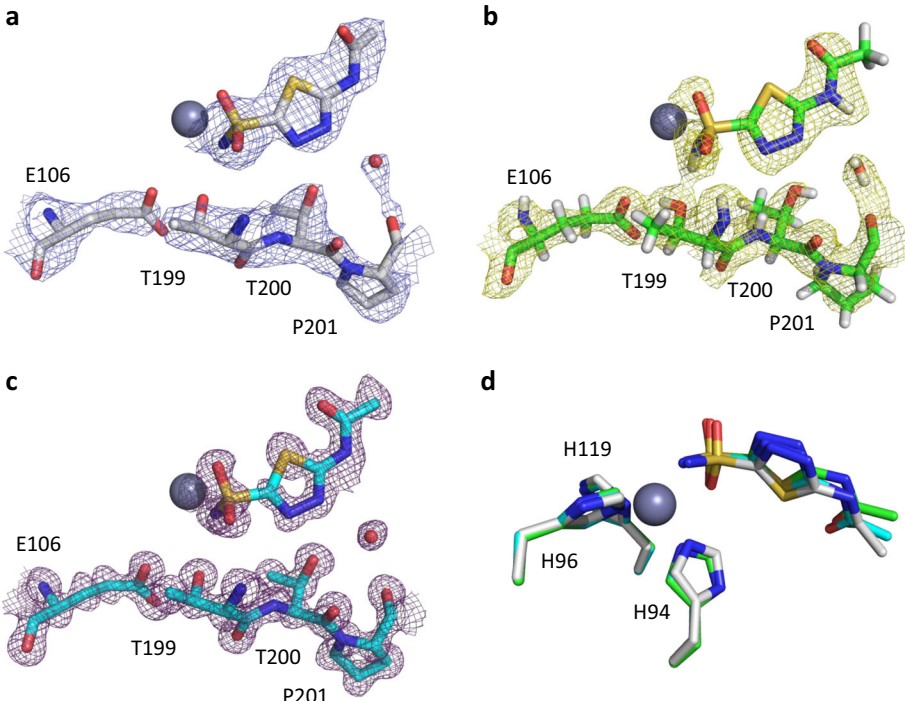

**Fig. 4 Model and map comparison of HCA II:AZM complex determined by electron, neutron, and X-ray diffraction. a** Electrostatic potential map from the MicroED data is shown in blue, contoured at $1.2\sigma$ (2.5 Å resolution, 80.0% completeness). **b** Nuclear-density map for joint refinement is shown in yellow, contoured at $1.2\sigma$ (PDB ID 4g0c, 2.0 Å resolution, 85.7% completeness). **c** Electron-density map is shown in magenta, contoured at $1.2\sigma$ (PDB ID 3hs4, 1.1 Å resolution, 94.7% completeness). **d** Overlay of the AZM ligand for the MicroED, neutron, and X-ray models. The models were aligned on the zinc coordination sphere, including the three active-site histidine residues. Carbon atoms are shown in gray, green, and cyan for the MicroED, neutron, and X-ray models, respectively. Hydrogen, nitrogen, oxygen, and sulfur atoms are colored in white, blue, red, and yellow, respectively. Zinc is shown as a dark-gray sphere.

may become a viable alternative in forthcoming studies for hit-to-lead optimization, revealing novel details of the underlying molecular mechanisms that mediate ligand binding to guide the design of novel drugs.

## Methods

**Protein expression and purification**. HCA II was expressed in BL21 DE3 pLysS *E. coli* cells that were transformed with an expression plasmid based on pET31F1 coding for protein with UniProt accession code P00918[45]. The cells were grown in shaker incubators at 180 rpm and 37 °C in LB media supplemented with 0.1 mg/ml ampicillin. Once the cells reached an optical density of ~1 at 600 nm, protein expression was induced with the addition of 1 mM isopropyl β-ᴅ-1-thiogalacto-pyranoside (IPTG) and 1 mM $ZnSO_4$. Protein expression was allowed to continue for 3 h. At this point, the cells were harvested by centrifugation in a JLA 8.1 rotor (Beckman) at ~6800 × $g$ for 20 min and frozen at −20 °C. Cells were resuspended in buffer A (0.2 M $Na_2SO_4$, 100 mM Tris, pH 9.0) and lysed by addition of lyso-zyme while stirring in the cold room for 3–4 h. The cell lysate was centrifuged in a JA-25.50 rotor (Beckman) at ~18,000 × $g$ for 60 min at 4 °C, and the supernatant containing the soluble cellular fraction was loaded onto affinity resin (*para*-ami-nomethylbenzensulfonamide, Sigma Aldrich A0796). Unbound protein and nucleic acids were removed by repeated wash steps with buffer A, followed by buffer B (0.2 M $Na_2SO_4$, 100 mM Tris, pH 7.0). Bound HCA II was eluted with 50 mM Tris, pH 7.8, and 0.4 M $NaN_3$. The eluted protein was concentrated with Amicon Ultra concentration devices (10 kDa molecular weight cutoff), purified, and buffer-exchanged by size-exclusion chromatography on HiLoad 26/600 Superdex 75 (GE Healthcare Life Sciences) in 50 mM Tris, pH 8.5. Fractions were collected and analyzed by sodium dodecyl sulfate-polyacrylamide gel electrophoresis to assess protein purity and homogeneity prior to crystallization. Eluted fractions containing HCA II were pooled and concentrated to ~20 mg/ml by using Amicon Ultra Centrifugal Filter Units (Merck) with a molecular weight cutoff of 10 kDa.

**Crystallization**. Crystals of HCA II were grown in sitting-drop vapor-diffusion setups using microbridges (Hampton Research) and 24-well Linbro crystallization plates (Hampton Research). The drops were prepared by mixing 10 μl of protein solution (20 mg/ml) with 10 μl of precipitant solution (2.8 M $(NH_4)_2SO_4$, 0.1 M Tris, pH 8.5), and were equilibrated against 1 ml of reservoir solution of the pre-cipitant. Plates were incubated at 20 °C, and crystals appeared in 2–3 days. Crystals

were harvested by crushing and repeated pipetting into 1.5 ml Eppendorf tubes to collect slurries of small crystals in mother liquor and stored at 20 °C until cryo-grid preparation.

**Sample preparation**. Complexes of HCA II with inhibitor AZM (Sigma Aldrich, A6011) were prepared by soaking the crystal slurries with inhibitor solubilized in dimethyl sulfoxide (DMSO) for 20 min, with a final concentration of 0.5 mM HCA II and 4.5 mM AZM. Grids were prepared by pipetting 3 μl on a QUANTIFOIL 1.2/1.3 (300 mesh) Cu holy carbon TEM grid. Excess liquid was removed via pressure-assisted backside blotting using Preassis[46]. The grid was vitrified manually by flash-cooling in liquid ethane. The sample was transferred to a Gatan 914 cryo-transfer holder.

**Data acquisition**. Microcrystal electron-diffraction data were collected on a JEOL JEM-2100 ($LaB_6$ filament) TEM operated at 200 kV equipped with a Timepix hybrid pixel detector (Amsterdam Scientific Instruments). Grids were screened for suitable microcrystals in defocused diffraction mode, and diffraction data were collected from an area of 1.5 μm diameter defined by a selected-area aperture using the *Instamatic* software interface[40]. The effective sample-to-detector distance was 1481.74 mm. Data were collected using continuous rotation with an angular increment of 0.68° and an exposure time of 1.5 s per frame. Individual crystal datasets were typically collected over a tilt range of on average 10–30°, with an acquisition time of 22–66 s per crystal dataset. The total range of tilt angles covered during data collection from several crystals was −60 to +60°. The electron dose rate applied during data collection was approximately 0.1 e⁻/Å²/s, and the total exposure dose for each crystal dataset was within 2.0–6.0 e⁻/Å².

**Data processing**. Data were integrated using XDS[47] with the Laue group con-strained to 2/*m*. Reflections from different crystal datasets were merged in XSCALE[47] using a weighted average of the unit-cell parameters obtained from individual crystal processing (see Supplementary Tables 1 and 2). Selection criteria for including data were based on merging statistics and correlation coefficients between individual datasets as reported by XSCALE. A cutoff value of 0.7 was used for the ligand-bound HCA II data; for the native HCA II data, a cutoff of 0.6 was used. Data were truncated at approximately $I/\sigma I \geq 1.0$ and $CC_{1/2} \geq 0.4$ with a correlation significant at the 0.1% level[48] (see Supplementary Tables 3 and 4). Data were merged and converted into MTZ format using AIMLESS[49].

**Structure solution**. A search model of the apo structure with the metal cofactor, ligands, and water removed was generated from a high-resolution X-ray model (PDB ID 3hs4[29]) using Sculptor[50]. The structure was solved using maximum-likelihood molecular replacement in Phaser[36] using the MicroED reflection intensities. For both native and ligand-bound data, a well-contrasting single solution was found with LLG = 1871, TFZ = 18.6, and LLG = 2494, TFZ = 19.6, respectively.

**Model building and refinements**. Starting electrostatic potential maps were generated from the molecular replacement solution using rigid-body refinement in *phenix.refine*[51]. The starting model and maps were used for remodeling several side chains and manually placing the metal cofactor $Zn^{2+}$ using Coot[37]. After restrained reciprocal space refinement in *phenix.refine*, the resulting map was inspected for any difference potential, indicating the presence of the bound inhibitor. The AZM ligand was imported in Coot using the get monomer command, and we used the find ligands command to fit the ligand using the precalculated mFo–DFc difference potential map at $2.8\sigma$.

Furthermore, upon inspection of the map, a clear blob of difference potential was observed in the solvent region, appropriate for accommodating a single DMSO molecule that was used as solvent to solubilize the ligand. The DMSO molecule was manually fitted using the get monomer command in Coot. Its position is in agreement with the neutron structure (PDB ID 4g0c)[30] where the space is occupied by three waters, and with a glycerol (GOL) solvent molecule occupying the same region in the high-resolution X-ray model (PDB ID 3hs4)[29].

The HCA II:AZM model was then completed, automatically placing water molecules, and using restrained reciprocal space refinement in *phenix.refine*[51]. All refinement steps were preformed using a test set representing 5% of all reflections, atomic scattering factors for electrons, automatic weighting of the experimental data to stereochemistry and atomic displacement parameter terms, and group *B*-factor refinement per residue[51].

**Validation**. The geometry of the native and ligand-bound structural models was validated using MolProbity[52] (Table 1). A simulated annealing (SA) composite omit map was calculated using *phenix.composite_omit_map*[51] by sequentially omitting 5% fractions of the structure. No missing reflections were filled in for map calculations. The MicroED model of the native structure shows a backbone Cα root-mean-square (r.m.s.) deviation of 0.23 Å with a structure from joint refinement against neutron and X-ray diffraction data (PBD ID 3tmj)[27], and 0.22 Å with an X-ray structure of native HCA II (PBD ID 3ks3)[53]. The inhibitor-bound MicroED model shows a backbone Cα r.m.s. deviation of 0.29 Å with a previously determined model from joint X-ray and neutron refinement (PDB ID 4g0c)[30], and of 0.22 Å with a high-resolution X-ray model of the same inhibitor-bound complex (PDB ID 3hs4)[29], that was used as search model for molecular replacement. Root-mean-square deviation values between structural models were calculated by the secondary-structure matching (SSM) tool[54].

**Figures**. Figures 2–4 were prepared using the PyMOL Molecular Graphics System, version 2.2.3 Schrödinger, LLC.

**Reporting summary**. Further information on research design is available in the Nature Research Reporting Summary linked to this article.

## Data availability
The atomic coordinates of ligand-bound and native HCA II are deposited in the Protein Data Bank under accession codes 6YMA and 6YMB, respectively. MicroED data of HCA II:AZM and native HCA II are available online at the SBGrid Data Bank under https://doi.org/10.15785/SBGRID/792 and https://doi.org/10.15785/SBGRID/793, respectively. All remaining data will be available from the corresponding author upon request.

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

## Acknowledgements
The project is supported by the Swedish Research Council (2017-05333, H.X.; 2019-00815, X.Z.), the Knut and Alice Wallenberg Foundation (2018.0237, X.Z.), and the Science for Life Laboratory through the technique development grant (Micro-ED@SciLifeLab, H.X.). Open access funding provided by Stockholm University.

## Author contributions
M.T.B.C. contributed to sample preparation, electron-diffraction data collection, data analysis, structure determination, paper writing, and figure making. S.Z.F. contributed to project design, crystal growth, sample preparation, and paper writing. M.C. contributed to the data analysis, X.Z. contributed to the conception and paper writing. H.X. contributed to the project design, conception, sample preparation, electron-diffraction data collection, data analysis, paper writing, and figure making.

## Competing interests
The authors declare no competing interests.
