## [Peer Review File · Communications Biology]

Reviewers' comments:

Reviewer #1 (Remarks to the Author):

My opinion on the paper 'Visualizing drug inhibitor binding interactions using microcrystal electron diffraction' is that the paper should be published as it is. The paper shows how electron diffraction can be successfully used to detect the binding of an inhibitor to a protein in a crystal of few microns a remarkable advantage for two reasons: i) less difficulty in crystallizing the protein ii) less difficulty in efficiently soaking the inhibitor in a smaller protein crystal.

I have just two minor remark:

1. I have seen that the distance between the sulfonamide N and the metal cofactor is more similar to the neutron than to the x-ray result. Can it be that since the x-ray feels the electron density they see a N displaced towards the Zn because of the electron lone pairs, an effect that it is not seen in neutron that feels the nuclear scattering? If this is the case it can be explained in a similar way with electron diffraction since the electrons feel the electrostatic potential which is positively peaked around the nucleus of the atoms....It would be interesting a discussion on that that can be used also to explain why electron see O-H larger than those obtained by x-ray.
2. In figure 3 I found some contradictions to what is written in the caption and what is displayed in the picture. In the caption the distance between the sulfonamide N and the Zn is reported as 2.0 while in the picture is reported 2.4. Again in the caption the hydrogen bonding between this N and the Thr199 is reported to be 2.6 while in the picture it is written 2.4.

Reviewer #2 (Remarks to the Author):

In this work the authors present a MicroED structure at sufficient resolution to clearly identify the interaction of an inhibitor with human carbonic anhydrase. There are a few points that should be addressed in the manuscript (given below):

-A major point is that while this is the first study where the MicroED data alone was sufficient to model the inhibitor, in the previous study on bevirimat cited by the authors (Purdy, et al, 2018) it is my understanding that the MicroED data were used along with other methods to identify the position of the inhibitor. Because the MicroED data were key to identifying the position of bevirimat, the statements on this being the first study should be qualified. Instances saying things such as "so far no drug inhibitor binding interactions could unambiguously be resolved by electron diffraction" (abstract), should be qualified to say that electron diffraction, or MicroED, "alone" has not been used.

-The angular space sampled per crystal was relatively small while still delivering a somewhat standard dose. Was the high dose per degree sampled for each crystal to improve resolution? the authors should briefly comment on this since it seems a little non-standard

-Line 318 states "...difference density was observed in the solvent region, too small to be water but appropriate for DMSO". this is unclear on what this means? water is smaller than DMSO, so what is exactly smaller?

Reviewer #3 (Remarks to the Author):

The paper entitled "Visualizing drug inhibitor binding interactions using microcrystal electron diffraction" by Clabbers, et al, presents the structure of both apo and inhibitor-bound HCA II, determined by microcrystal electron diffraction (microED). The inhibitor the authors selected, acetazolamide, is a clinical drug, and the authors use this case study to support the claim that

microED is a useful tool for drug discovery. I agree with the authors that microED (currently a bit of a hammer without a nail in the absence of methods for experimental phasing) is quite likely to find utility in drug discovery. The reasons are highlighted by the authors – small crystals soak more easily due to their small volumes, which result in rapid diffusion times and reduction of stress caused by soaking. Additionally, the ability to screen crystals quickly by microED is under development and will no doubt be possible soon. All of these benefits make the technique attractive to pharmaceutical companies that can easily afford the modest electron microscopes required to perform the method in-house (it could be worth mentioning this point). For these reasons, I feel that the case study presented by the authors would be appropriate for publication in *Communications Biology*, and I believe that as the technique grows this early example is likely to receive numerous citations. I do, however have a few technical concerns, as well as some other minor comments, and I suggest some revisions be performed prior to publication.

My biggest concern is that the authors claim that microED has utility in drug discovery (I agree), and they comment, in the first sentence of the abstract: “Visualizing drug inhibitor binding interactions at the atomic level is important for both structure-based drug design and fragment-based screening methods.” This statement is a dramatic oversimplification given their claim. It seems important to be a little more nuanced about precision throughout the manuscript, and be clearer about where microED might fit into a drug discovery pipeline. For example, with fragment screening approaches, where the goal is to identify novel chemical matter and/or binding surfaces, resolution and precision might be less important than throughput. On the other hand, hit-to-lead optimization often requires high resolution and precision on the order of 100s of picometers. My hunch, though I’m not sure, is that microED is more likely to be useful for the former, and not the latter. The authors could provide some clarity here both visually and numerically, by doing one or more of the following:

- In Fig. 4, it would be useful to show a “panel D” with all of the bound inhibitor poses overlaid, to show if the refined ligand position differs at all between the X-ray, neutron, and microED structures. The key will be doing the alignment in a sensible way. Rather than doing a global alignment, I suggest aligning the structures on a local and well resolved feature, such as the Zn atom and its coordination sphere, and then evaluating the position of the inhibitor.
- Another way of doing this independently of the alignment would be to calculate the pairwise distance between each of the AZM atoms and the Zn atom for each of the three structures and compare those distances.

I also think it is critical to provide a measure of the coordinate precision in the models. A simple way to do this is to look in the phenix.refine log files for the maximum-likelihood estimate of coordinate precision. For X-ray structures this tends to be about 1/10 of the resolution for a “good structure.” I’m not aware of anyone reporting these metrics for microED, and it seems relevant here. A more rigorous, but also more time consuming way of assessing this is empirically, by refining the same structure against the same data ≥ 10 times, each with different random seeds and coordinate perturbations, and then looking at the RMSD of atom positions across all of the parallel refinements.

Next, I have some concerns about the crystallographic data.

1. Are there multiple lattices recorded in the diffraction images? To my eye, Fig 1D appears to have multiple incompatible lattice lines evident in the image. Was there any attempt to confirm that the data processing was handling this appropriately and choosing reflections from a single lattice (e.g. by overlaying the predictions and the observations)?

2. Since the data were merged from many crystals, it would be nice to see some information showing the distribution of the observed unit cell parameters. This is important given that very small instances of non-isomorphism can cause large incompatibilities in measured intensities (see Crick and Magdoff (1956) for theoretical treatment). Also, were any criteria used to assess whether individual data sets should be included in the merge, such as behavior of overall statistics upon inclusion/exclusion?

3. The authors' claim of 73% completeness, already extremely low, is somewhat misleading, since the high-resolution data are closer to 50% complete. In the case of resolving features with high precision, as one would want for drug discovery, the high-resolution data are carrying much of the useful information. This problem is compounded by the fact that the data are not just incomplete, but there is orientation bias, which means that the information content of the data set is anisotropic. I don't have any issue with the $I/\sigma > 1.0$ and $CC1/2 > 0.3$ criteria used by the authors to set their resolution cutoff, but completeness needs to be accounted for. In this case, it seems like it would be prudent to reduce the resolution of the data to increase the completeness in the high-resolution bins. I expect this would also lead to improved R/Rfree. I understand that completeness can be challenging in microED, but at a minimum I feel that this potential limitation needs to be addressed more directly in the manuscript.

4. I'm a bit confused by the B-factors reported by the authors, both in Table 1 and in Fig. 3b. In Table 1, the authors report that for both of their structures, the ligand/ion and solvent B factors are substantially (by as much as a factor of two) lower than the B-factors for the protein atoms. This is highly unusual, and can be a red flag for something being incorrect about the refinement and/or model. Physically, it just doesn't make sense for a protein molecule to be more disordered than the solvent and ligands that are bound to it. Indeed, when judging the refinement of ligand B-factors and occupancies, one often checks to ensure the ligand B-factors are approximately equal to those of the protein atoms that surround it. The appearance of the colors in Fig. 3b make more intuitive sense to me, in terms of how I would expect the ligand Bfactors to behave, but those colors don't appear to align with the averages reported in the table. Are there some extremely poorly resolved parts of the protein with really high B-factors that are causing this incongruity? If so, is there part of the model that needs to be reevaluated?

I have the following minor comments:

- The authors comment, in multiple places, on rapid soaking times being a major benefit of small crystals that can be measured by microED, and they describe in their methods that they used a "brief 20 minute soak." Personally, I wouldn't consider 20 minutes to be a "brief soak." In many cases, 20 minutes is more than sufficient to soak ligands into crystals that are hundreds of microns in each dimension, and in the field of time-resolved crystallography, sub-ms mixing times have been reported for crystals smaller than a micron. Why did the authors choose to use such a long soak? Was this length of soaking required? If so, it somewhat subverts the claim about rapid soaking times.
- The phrase "drug inhibitor" is used in several places to refer to the AZM molecule. AZM is a drug, and it is an inhibitor, but putting these two words together and calling it a "drug inhibitor" is unusual and somewhat awkward. (To me, "drug inhibitor" sounds like something that inhibits the activity of a drug.) I'd personally just call it a "drug," in order to emphasize that it is a clinical molecule, or call it an "inhibitory drug" if you want to be specific about MOA.
- In the first paragraph of the introduction, the authors comment on the use of thin microcrystals. Since this paper is likely to attract an audience with drug discovery backgrounds who are unfamiliar with microED in general, it could be useful to quantify exactly how small the crystals are.
- On line 67, I suggest replacing the word "inhibitor" with "small molecule." They are rarer, but

drugs can also be activators.

- The reference on line 69 seems like it could also include some of the earlier examples of crystallographic fragment screening, in addition to this more recent paper (which is also very nice).
- The authors make a couple of references to another microED study of drug binding; however they erroneously refer to these studies as focusing on HIV protease. The paper in question (ref 19) is a study of a drug binding to the CTD of the HIV Gag protein. The drug does block proteolysis, but it does so by trapping the substrate in a conformation that is inaccessible to the protease. The drug, bevirimat, does not interact directly with the protease. The authors are correct that the orientation of this small molecule could not be unambiguously resolved in that study, however it seems a bit unfair to make that point without also mentioning that the binding site lies on a symmetry axis at the center of a homo-multimer.
- The authors give a beautiful description of carbonic anhydrase chemistry and reaction mechanism, and also comment that it is a drug target, but there is little connection between those ideas. Can the authors provide a sentence or two to put the chemistry in the context of the pathology (glaucoma, altitude sickness, congestive heart failure). Why does blocking this reaction help those conditions?
- Description of the structure refinement in lines 148-149 seems inconsistent with the methods section. The final refinement step was a reciprocal space refinement, right? That's what is described in the methods.

As stated, I believe this paper adds value to the rapidly-growing field of microED by demonstrating a viable use case for the maturing technique, and will also capture the interest of the drug discovery community by describing an attractive pipeline for in-house structure determination to support small-molecule development. I support its publication, provided the authors can address the above concerns. Finally, I commend the authors for posting their manuscript as a preprint simultaneous with peer-review, and I urge them to also check any comments on the preprint server that might improve the quality of the manuscript.

I review non-anonymously,
Michael Thompson
Assistant Professor, UC Merced

Response letter to COMMSBIO-20-1031-T

Please note that we have uploaded a revised version of our manuscript entitled "Visualizing drug binding interactions using microcrystal electron diffraction" (COMMSBIO-20-1031-T) in light of reviewer comments on the original submission. We thank the reviewers for their interest. Their constructive comments and questions helped us to clarify some parts of our manuscript and led to a better presentation of the work.

Here we reply to each of the review comments in turn and describe the changes we have made to accommodate them. The original review comments are shown in black, our response is shown in blue. In addition, we highlighted all changes in yellow in the revised manuscript.

Reviewer #1 (Remarks to the Author):

My opinion on the paper 'Visualizing drug inhibitor binding interactions using microcrystal electron diffraction' is that the paper should be published as it is. The paper shows how electron diffraction can be successfully used to detect the binding of an inhibitor to a protein in a crystal of few microns a remarkable advantage for two reasons: i) less difficulty in crystallizing the protein ii) less difficulty in efficiently soaking the inhibitor in a smaller protein crystal.

I have just two minor remark:

1. I have seen that the distance between the sulfonamide N and the metal cofactor is more similar to the neutron than to the x-ray result. Can it be that since the x-ray feels the electron density they see a N displaced towards the Zn because of the electron lone pairs, an effect that it is not seen in neutron that feels the nuclear scattering? If this is the case it can be explained in a similar way with electron diffraction since the electrons feel the electrostatic potential which is positively peaked around the nucleus of the atoms...It would be interesting a discussion on that that can be used also to explain why electron see O-H larger than those obtained by x-ray.

Response: Electrons see the electrostatic potential and are scattered by both the nuclei and electron clouds. Electrons do have improved contrast for visualizing hydrogen atoms compared to x-ray diffraction because of the electron form factors (see for example Palatinus *et al.*, *Science* 355, 166-169 (2017), and Clabbers *et al.*, *Acta Cryst.* A75, 82.93 (2019)). As a result, the hydrogen-bond lengths we observe in electron diffraction (for small molecular compounds) are indeed larger than observed by x-ray diffraction, and are closer to the more accurate hydrogen-bond lengths for neutron scattering.

We have added the following sentence to the last paragraph of the discussion, line 254: *"Furthermore, electrons have improved contrast for visualizing hydrogen atoms compared to x-ray diffraction, and observed hydrogen-bond lengths are closer to the more accurate inter-nuclei distances for neutron scattering^{42,43}."*

The two added references 42 and 43 refer to the following publications:

42. Palatinus, L. *et al.* Hydrogen positions in single nanocrystals revealed by electron diffraction. *Science* **355**, 166–169 (2017).
43. Clabbers, M. T. B., Gruene, T., van Genderen, E. & Abrahams, J. P. Reducing dynamical electron scattering reveals hydrogen atoms. *Acta Cryst. A.* **75**, 82–93 (2019).

2. In figure 3 I found some contradictions to what is written in the caption and what is displayed in the picture. In the caption the distance between the sulfonamide N and the Zn is reported as 2.0 while in the picture is reported 2.4. Again in the caption the hydrogen bonding between this N and the Thr199 is reported to be 2.6 while in the picture it is written 2.4.

Response: We have corrected this in the revised manuscript. The values given in Figure 3 and its caption are now consistent. In response to one of the comments raised by reviewer 3, we refined the model of the ligand-bound protein against data truncated at 2.5 Å. We updated Figure 3 with the maps truncated at 2.5 Å resolution. After structure determination and ligand fitting against the truncated data, the distance of the sulfonamide nitrogen to the zinc atom is 2.1 Å, and that of the nitrogen to the oxygen of THR199 is 2.6 Å.

Please find the revised Figure 3 enclosed at the end of the response letter.

Reviewer #2 (Remarks to the Author):

In this work the authors present a MicroED structure at sufficient resolution to clearly identify the interaction of an inhibitor with human carbonic anhydrase. There are a few points that should be addressed in the manuscript (given below):

-A major point is that while this is the first study where the MicroED data alone was sufficient to model the inhibitor, in the previous study on bevirimat cited by the authors (Purdy, et al, 2018) it is my understanding that the MicroED data were used along with other methods to identify the position of the inhibitor. Because the MicroED data were key to identifying the position of bevirimat, the statements on this being the first study should be qualified. Instances saying things such as "so far no drug inhibitor binding interactions could unambiguously be resolved by electron diffraction" (abstract), should be qualified to say that electron diffraction, or MicroED, "alone" has not been used.

Response: We changed the manuscript accordingly. Line 19 of the abstract now reads: "*However, so far no drug binding interactions could unambiguously be resolved by electron diffraction alone.*" We changed line 71 of the introduction to: "*A unique binding pose could however not be resolved from the MicroED data alone*".

-The angular space sampled per crystal was relatively small while still delivering a somewhat standard dose. Was the high dose per degree sampled for each crystal to improve resolution? the authors should briefly comment on this since it seems a little non-standard

Response: We indeed used a relatively high dose over a small angular range in an attempt to improve the resolution of the data. We have added a comment regarding our dose rate to the results section, line 130: *“In an attempt to optimize the signal-to-noise ratio and improve the resolution, we collected data over small angular ranges of on average 10-30° with a relatively high dose rate of 0.1 e⁻/Å²/s. The accumulated electron dose was typically within 2.0-6.0 e⁻/Å² per dataset in order to minimize radiation damage^{8,35}.”*

-Line 318 states "...difference density was observed in the solvent region, too small to be water but appropriate for DMSO". this is unclear on what this means? water is smaller than DMSO, so what is exactly smaller?

Response: We meant to state that a single water molecule is smaller and would not explain the observed difference potential, whereas the difference potential seemed appropriate to accommodate the larger DMSO molecule. We changed the statement in line 344 to *“Furthermore, upon inspection of the map, a clear blob of difference potential was observed in the solvent region, appropriate for accommodating a single DMSO molecule that was used as solvent to solubilize the ligand. The DMSO molecule was manually fitted using the get monomer command in Coot.”*

Reviewer #3 (Remarks to the Author):

The paper entitled “Visualizing drug inhibitor binding interactions using microcrystal electron diffraction” by Clabbers, et al, presents the structure of both apo and inhibitor-bound HCA II, determined by microcrystal electron diffraction (microED). The inhibitor the authors selected, acetazolamide, is a clinical drug, and the authors use this case study to support the claim that microED is a useful tool for drug discovery. I agree with the authors that microED (currently a bit of a hammer without a nail in the absence of methods for experimental phasing) is quite likely to find utility in drug discovery. The reasons are highlighted by the authors – small crystals soak more easily due to their small volumes, which result in rapid diffusion times and reduction of stress caused by soaking. Additionally, the ability to screen crystals quickly by microED is under development and will no doubt be possible soon. All of these benefits make the technique attractive to pharmaceutical companies that can easily afford the modest electron microscopes required to perform the method in-house (it could be worth mentioning this point). For these reasons, I feel that the case study presented by the authors would be appropriate for publication in Communications Biology, and I believe that as the technique grows this early example is likely to receive numerous citations. I do, however have a few technical concerns, as well as some other minor comments, and I suggest some revisions be performed prior to publication.

Response: We would like to thank the reviewer for taking the time and effort to write a very extensive and constructive review. Please find our response to the comments below.

My biggest concern is that the authors claim that microED has utility in drug discovery (I agree), and they comment, in the first sentence of the abstract: “Visualizing drug inhibitor

binding interactions at the atomic level is important for both structure-based drug design and fragment-based screening methods.” This statement is a dramatic oversimplification given their claim. It seems important to be a little more nuanced about precision throughout the manuscript, and be clearer about where microED might fit into a drug discovery pipeline. For example, with fragment screening approaches, where the goal is to identify novel chemical matter and/or binding surfaces, resolution and precision might be less important than throughput. On the other hand, hit-to-lead optimization often requires high resolution and precision on the order of 100s of picometers. My hunch, though I’m not sure, is that microED is more likely to be useful for the former, and not the latter.

Response: We have adapted the manuscript accordingly, rephrasing the narrative more towards highlighting the opportunities for fragment-based screening using MicroED. Future improvements in instrumentation and data collection strategies may push the resolution and improve both data accuracy and precision, making MicroED potentially also viable for structure-guided drug design.

We changed the passages in the manuscript that discuss the potential of MicroED for drug discovery experiments.

The abstract now reads:

“Visualizing ligand binding interactions is important for structure-based drug design and fragment-based screening methods. Rapid and uniform soaking with potentially reduced lattice defects make small macromolecular crystals attractive targets for studying drug binding using microcrystal electron diffraction (MicroED). However, so far no drug binding interactions could unambiguously be resolved by electron diffraction alone. Here, we use MicroED to study the binding of a sulfonamide inhibitor to human carbonic anhydrase isoform II (HCA II). We show that MicroED data can efficiently be collected on a conventional TEM from thin hydrated microcrystals soaked with the clinical drug acetazolamide (AZM). The data are of high enough quality to unequivocally fit and resolve the bound inhibitor. We anticipate MicroED can play an important role in facilitating in-house fragment screening for drug discovery, complementing existing methods in structural biology such as x-ray and neutron diffraction.”

We rephrased the final paragraph (lines 250 to 261) of the discussion in the following manner:

“We compare our MicroED data against electron and nuclear density maps obtained by x-ray and neutron diffraction, respectively (Fig. 4). Our electrostatic potential map is well-resolved despite limited resolution and low data completeness. Electrons offer certain advantages, for example the possibility to refine atomic charges, exploring the charged state of protein-ligand binding interactions^{10,41}. Furthermore, electrons have improved contrast for visualizing hydrogen atoms compared to x-ray diffraction, and hydrogen-bond lengths are closer to the more accurate inter-nuclei distances for neutron scattering^{42,43}. With additional improvements in instrumentation and data collection strategies, resolution and data completeness can be improved, and increase the accuracy and precision of electron diffraction data^{35,44}. Given these prospect, MicroED may become a viable alternative in

forthcoming studies for hit-to-lead optimization revealing novel details of the underlying molecular mechanisms that mediate ligand binding to guide the design of novel drugs.”

The authors could provide some clarity here both visually and numerically, by doing one or more of the following:

- In Fig. 4, it would be useful to show a “panel D” with all of the bound inhibitor poses overlaid, to show if the refined ligand position differs at all between the X-ray, neutron, and microED structures. The key will be doing the alignment in a sensible way. Rather than doing a global alignment, I suggest aligning the structures on a local and well resolved feature, such as the Zn atom and its coordination sphere, and then evaluating the position of the inhibitor.
- Another way of doing this independently of the alignment would be to calculate the pairwise distance between each of the AZM atoms and the Zn atom for each of the three structures and compare those distances.

Response: We have added a fourth panel to Figure 4 with superimposed inhibitor-bound protein active sites for our MicroED model and previously determined x-ray (PDB ID 3hs4) and neutron (PDB ID 4g0c) diffraction structures of the same HCA II:AZM complex. We aligned the three models to the zinc atom and the three histidine residues it is coordinated to. We calculated the RMSD of the atomic coordinates of the ligand against the overlaid x-ray and neutron models using the superimposed coordinate matrix. The RMSD calculated over the 14 atoms of the ligand is 0.466 Å with respect to the x-ray model, and 0.679 Å between the MicroED and neutron model. The RMSD of the ligand atomic coordinates between the x-ray and neutron models is 0.409 Å.

We updated panels A to C to have the same colour coding of the carbon backbone as in panel 1D to distinguish between the MicroED, x-ray, and neutron models. As we compare the x-ray (PDB ID 3hs4) and neutron (PDB ID 4g0c) models in 4D, we show the x-ray diffraction data of 3hs4 in panel 4C. Previously, we presented the x-ray data used in joint refinement of 4g0c. The electrostatic scattering potential map shown in 4A was calculated for the MicroED truncated at 2.5 Å resolution, as response to question 4. Please refer to our response to question 4 for a more detailed description.

We changed the caption of Figure 4 accordingly. It’s header was changed to: “*Model and map comparison of HCA II:AZM complex determined by electron, neutron, and x-ray diffraction.*”. The caption added for panel 4D is: “*Overlay of the AZM ligand for the MicroED, neutron, and x-ray models. The models were aligned on the zinc coordination sphere including the three active site histidine residues. Carbon atoms are shown in grey, green and cyan for the MicroED, neutron, and x-ray models, respectively.*”

The updated Figure 4 can be found at the end of the response letter.

In the results section, we added the following passage, discussing the comparison of the aligned ligand binding poses, starting from line 214: “*To compare the binding poses of the ligand, we aligned the MicroED, neutron, and x-ray models with respect to the zinc atom and*

its coordination sphere (Fig. 4d). We calculated the r.m.s. deviation in atomic coordinates of the AZM ligand (14 atoms) aligned with respect to the zinc atom. We found a r.m.s. deviation of 0.466 Å for the x-ray model with respect to the MicroED structure, and 0.679 Å for neutron model. Between the x-ray and neutron model, the r.m.s. deviation in atomic coordinates is 0.409 Å.

I also think it is critical to provide a measure of the coordinate precision in the models. A simple way to do this is to look in the phenix.refine log files for the maximum-likelihood estimate of coordinate precision. For X-ray structures this tends to be about 1/10 of the resolution for a “good structure.” I’m not aware of anyone reporting these metrics for microED, and it seems relevant here. A more rigorous, but also more time consuming way of assessing this is empirically, by refining the same structure against the same data ≥ 10 times, each with different random seeds and coordinate perturbations, and then looking at the RMSD of atom positions across all of the parallel refinements.

Response: We have added the maximum-likelihood estimates of coordinate precision to the results section. The coordinate precision is 0.41 Å for the native structure, and 0.37 Å for the inhibitor-bound protein model.

We added the following two sentences to the results section, line 155: “*The final structural model has a Rwork/Rfree of 0.249/0.276 (Table 1), and a coordinate precision (maximum-likelihood estimate) of 0.41 Å.*”, and line 164: “*The model has a final Rwork/Rfree of 0.224/0.255 (Table 1), and a reported coordinate precision (maximum-likelihood estimate) of 0.37 Å.*”

Furthermore, as suggested we refined the structure 10 times using each time a different random seed with shaking of the atomic coordinates by random perturbations of 0.3Å. This resulted in a core RMSD of the protein backbone of 0.052 Å, and a RMSD of 0.073 Å for the ligand.

We added the following sentences to the results section, starting at line 166: “*To further assess the coordinate precision of our model, we refined the structure 10 times using each time a different random seed with shaking of the atomic coordinates by random perturbations of 0.3 Å. This resulted in a core root-mean-squares (r.m.s.) deviation of the atomic coordinates for the parallel refinements of 0.052 Å for the protein backbone (257 residues), and a r.m.s. deviation of 0.073 Å for the inhibitor (14 atoms).*”

Next, I have some concerns about the crystallographic data.

1. Are there multiple lattices recorded in the diffraction images? To my eye, Fig 1D appears to have multiple incompatible lattice lines evident in the image. Was there any attempt to confirm that the data processing was handling this appropriately and choosing reflections from a single lattice (e.g. by overlaying the predictions and the observations)?

Response: We did confirm that data processing was handled correctly and only integrated reflections of a single lattice by evaluating predicted spot positions. We show in the figure below (Figure 1, predicted spot positions from FRAME.cbf of the diffraction pattern shown in

Figure 1D) that XDS was able to predict the positions of the reflections in the electron diffraction frame based on the orientation matrix, suggesting that the lattice belongs to a single crystal.

Figure 1: FRAME.cbf

Furthermore, we have uploaded the raw electron diffraction to the SBGrid Data Bank to become publically available after publication of the manuscript. We added the doi for accessing the data under the paragraph Data availability, line 520: “*MicroED data of HCA II:AZM and native HCA II are available online at the SBGrid Data Bank under doi:10.15785/SBGRID/792 and doi:10.15785/SBGRID/793, respectively.*”

2. Since the data were merged from many crystals, it would be nice to see some information showing the distribution of the observed unit cell parameters. This is important given that very small instances of non-isomorphism can cause large incompatibilities in measured intensities (see Crick and Magdoff (1956) for theoretical treatment). Also, were any criteria used to assess whether individual data sets should be included in the merge, such as behavior of overall statistics upon inclusion/exclusion?

Response: We have added two supplementary tables with individual unit cell parameters and estimated standard deviations for each crystal (Supplementary Table 1 contains the

parameters for the native data, Supplementary Table 2 shows the dimensions for the ligand-bound crystals). Selection criteria for including individual datasets for merging were made based on overall merging statistics and correlation coefficients between datasets. Individual datasets with poor intensity statistics or highly diverging unit cell parameters were also excluded. For data merging we took the weighted average of all crystals used for merging, and we present these dimensions with the estimated standard deviations in Table 1. Furthermore, we included the detailed merging statistics per resolution bin as reported by XSCALE as supplementary information (Supplementary Table 3 shows the merging statistics for the native data, Supplementary Table 4 contains the statistics reported for the ligand-bound protein).

We updated the paragraph on the MicroED data processing in the methods section to include more detailed information with respect to data processing and merging. The entire paragraph (lines 135-144) now reads: “**Data processing.** Data were integrated using XDS⁴⁷ with the Laue group constrained to 2/m. Reflections from different crystal datasets were merged in XSCALE⁴⁷ using a weighted average of the unit cell parameters obtained from individual crystal processing (see Supplementary Table 1 and 2). Selection criteria for including data were based on merging statistics and correlation coefficients between individual datasets as reported by XSCALE. A cutoff value of 0.7 was used for the ligand-bound HCA II data, for native HCA II a cutoff of 0.6 was used. Data were truncated at approximately $I / \sigma I \geq 1.0$ and $CC_{1/2} \geq 0.4$ with a correlation significant and the 0.1% level⁴⁸ (see Supplementary Table 3 and 4). Data were merged and converted to MTZ format using AIMLESS⁴⁹.”

3. The authors’ claim of 73% completeness, already extremely low, is somewhat misleading, since the high-resolution data are closer to 50% complete. In the case of resolving features with high precision, as one would want for drug discovery, the high-resolution data are carrying much of the useful information. This problem is compounded by the fact that the data are not just incomplete, but there is orientation bias, which means that the information content of the data set is anisotropic. I don’t have any issue with the $I/\sigma I > 1.0$ and $CC_{1/2} > 0.3$ criteria used by the authors to set their resolution cutoff, but completeness needs to be accounted for. In this case, it seems like it would be prudent to reduce the resolution of the data to increase the completeness in the high-resolution bins. I expect this would also lead to improved R/Rfree. I understand that completeness can be challenging in microED, but at a minimum I feel that this potential limitation needs to be addressed more directly in the manuscript.

Response: We re-assessed our data with respect to the data completeness. Owing to preferred orientation and limited tilt range of the goniometer we indeed have an orientation bias. This does affect the map quality and generally leads to poorer resolved features in the direction of the missing information. The drop in completeness for the high resolution reflections can be attributed to the fact that not all crystal datasets diffracted up to 2.3Å, and because of the limited dimensions of our detector several high-resolution reflections would extend beyond the edge of the detector.

We truncated the HCA II:AZM data to 2.5Å to increase the completeness of the higher resolution bins. We added two supplementary tables with the merging statistics listed per

resolution bin (Supplementary Tables 3 and 4 for the native and ligand-bound data, respectively).

We discuss the data completeness in the results section, lines 319-327: *“Data processing. Data were integrated and merged to 2.5 Å resolution for structure determination, covering a combined angular range of -60 to +60° (Table 1, see Supplementary Table 1 and 2). The weighted average of the unit cell parameters are close to values described in literature^{29,30}. Owing to preferred crystal orientation, and limited tilt range of the TEM goniometer stage, overall data completeness did not increase beyond about 73% and 80% for the native and ligand bound protein, respectively (Table 1). The data are anisotropic predominantly along the direction of c* because of orientation bias. Completeness decreases in the lowest and highest resolution bins owing to shading of the reflections by the direct beam and varying crystallinity and diffracted intensity for different crystals, respectively (see Supplementary Table 3 and 4).”*

Truncating the data, and refining the structural models using group B-factors (as response to comment 4, please see for our detailed comments below), the data collection and refinement statistics are reported in Table 1 were affected. We updated the values in Table 1 accordingly. Furthermore, Figures 2, 3 and 4 were updated showing the maps and models for the truncated data at 2.5 Å resolution. Please see the end on the response letter for the updated figures. We have re-uploaded the latest coordinates to the PDB database under the same PDB codes.

4. I'm a bit confused by the B-factors reported by the authors, both in Table 1 and in Fig. 3b. In Table 1, the authors report that for both of their structures, the ligand/ion and solvent B factors are substantially (by as much as a factor of two) lower than the B-factors for the protein atoms. This is highly unusual, and can be a red flag for something being incorrect about the refinement and/or model. Physically, it just doesn't make sense for a protein molecule to be more disordered than the solvent and ligands that are bound to it. Indeed, when judging the refinement of ligand B-factors and occupancies, one often checks to ensure the ligand B-factors are approximately equal to those of the protein atoms that surround it. The appearance of the colors in Fig. 3b make more intuitive sense to me, in terms of how I would expect the ligand Bfactors to behave, but those colors don't appear to align with the averages reported in the table. Are there some extremely poorly resolved parts of the protein with really high B-factors that are causing this incongruity? If so, is there part of the model that needs to be reevaluated?

Response: We re-assed our data and several B-factors were indeed very low or relatively high. A refinement protocol using group B-factors per residues instead of individual B-factor refinement seemed more appropriate for the data resolution and completeness. Furthermore, the data were truncated to 2.5 Å resolution in response to comment 3. The updated B-factor values were added to the refinement statistics in Table 1. The reported mean B-factors are 20.65 Å² for the protein, 20.59 Å² for the waters, 20.58 Å² for the ligand.

We updated the methods sections with the different refinement protocol used (: *“The HCA II:AZM model was then completed, automatically placing water molecules, and using restrained reciprocal space refinement phenix.refine⁵¹. All refinement steps were performed*

*using a test set representing 5% of all reflections, atomic scattering factors for electrons, automatic weighting of the experimental data to stereochemistry and atomic displacement parameter terms, and group B-factor refinement per residue*⁵¹.”

We omitted Figure 3B, showing the *B*-factors per atom for the residues in the protein active site and the bound ligand. As we refined the structure using group B-factors this plot seemed no longer very relevant to be included. The *B*-factors of the ligand are in the same range as that of the protein. We modified line 191 of the results section to: *”The B-factors for the ligand are within the same range as those of the more stable side chains of the active site residues (Table 1).”*

I have the following minor comments:

- The authors comment, in multiple places, on rapid soaking times being a major benefit of small crystals that can be measured by microED, and they describe in their methods that they used a “brief 20 minute soak.” Personally, I wouldn’t consider 20 minutes to be a “brief soak.” In many cases, 20 minutes is more than sufficient to soak ligands into crystals that are hundreds of microns in each dimension, and in the field of time-resolved crystallography, sub-ms mixing times have been reported for crystals smaller than a micron. Why did the authors choose to use such a long soak? Was this length of soaking required? If so, it somewhat subverts the claim about rapid soaking times.

Response: We refrained from referring to the 20 minutes soak as fast or rapid in the revised manuscript. Ligand soaking is likely indeed already effective at much shorter time. We did not attempt to quantify or optimise soaking time.

- The phrase “drug inhibitor” is used in several places to refer to the AZM molecule. AZM is a drug, and it is an inhibitor, but putting these two words together and calling it a “drug inhibitor” is unusual and somewhat awkward. (To me, “drug inhibitor” sounds like something that inhibits the activity of a drug.) I’d personally just call it a “drug,” in order to emphasize that it is a clinical molecule, or call it an “inhibitory drug” if you want to be specific about MOA.

Response: We removed all instances of the combined use of “drug inhibitor” throughout the manuscript and now either refer to the AZM molecule as “drug”, “inhibitor”, or “ligand”. This also includes removing the word inhibitor from the title of our manuscript: *“Visualizing drug binding interactions using microcrystal electron diffraction”*

- In the first paragraph of the introduction, the authors comment on the use of thin microcrystals. Since this paper is likely to attract an audience with drug discovery backgrounds who are unfamiliar with microED in general, it could be useful to quantify exactly how small the crystals are.

Response: Crystal size for MicroED should ideally not exceed more than 500 nm thickness in the direction along the incident beam. We added the following statements quantifying the size

if the microcrystals we used, line 128: “*Microcrystals selected for data collection typically did not exceed a thickness of 500 nm, and diffracted beyond 2.5Å resolution (Fig. 1).*”

- On line 67, I suggest replacing the word “inhibitor” with “small molecule.” They are rarer, but drugs can also be activators.

Response: We changed the text accordingly replacing “inhibitor” with “small molecule”.

- The reference on line 69 seems like it could also include some of the earlier examples of crystallographic fragment screening, in addition to this more recent paper (which is also very nice).

Response: We added 3 additional references in addition to Collins *et al.*:

18. Kuhn, P., Wilson, K., Patch, M. G. & Stevens, R. C. The genesis of high-throughput structure-based drug discovery using protein crystallography. *Curr. Opin. Chem. Biol.* **6**, 704–710 (2002).
19. Hajduk, P. J. & Greer, J. A decade of fragment-based drug design: Strategic advances and lessons learned. *Nat. Rev. Drug Discov.* **6**, 211–219 (2007).
20. Blundell, T. L. Protein crystallography and drug discovery: Recollections of knowledge exchange between academia and industry. *IUCrJ* **4**, 308–321 (2017).

- The authors make a couple of references to another microED study of drug binding; however they erroneously refer to these studies as focusing on HIV protease. The paper in question (ref 19) is a study of a drug binding to the CTD of the HIV Gag protein. The drug does block proteolysis, but it does so by trapping the substrate in a conformation that is inaccessible to the protease. The drug, bevirimat, does not interact directly with the protease. The authors are correct that the orientation of this small molecule could not be unambiguously resolved in that study, however it seems a bit unfair to make that point without also mentioning that the binding site lies on a symmetry axis at the center of a homo-multimer.

Response: We revised our description of the results presented in the study by Purdy *et al.* accordingly. The passage in the manuscript (lines 68-73) now reads: “*The potential of MicroED for drug discovery was first indicated in recent work studying the binding of the inhibitor bevirimat (BVM) to the C-terminal domain of the HIV Gag protein²². The structural model provided insight into the underlying interactions in inhibitor binding. A unique binding pose could however not be resolved from the MicroED data alone. This was further complicated by the fact that the binding site is located on a symmetry axis at the center of the homo-multimer²².*”

- The authors give a beautiful description of carbonic anhydrase chemistry and reaction mechanism, and also comment that it is a drug target, but there is little connection between those ideas. Can the authors provide a sentence or two to put the chemistry in the context of the pathology (glaucoma, altitude sickness, congestive heart failure). Why does blocking this reaction help those conditions?

Response: We have added the following description to the introduction (lines 103-110): “Systemically, inhibition of extra- and intracellular CAs in the kidney causes increased excretion of bicarbonate, other ions (Na^+ , K^+ , Cl^-), and water through the urine. This leads to a mild diuretic effect and general metabolic acidosis with associated alkalization of urine³³. For the treatment of glaucoma, CA inhibitors are applied as topical eyedrops. Here inhibitors block CA in the ciliary epithelial cells, disrupting ionic transport across cell membranes, leading to reduced bicarbonate, Na^+ , and water transport across epithelium cells into the eye. The net effect is that less aqueous humor is formed, reducing intralocular pressure³⁴.”

References 33 and 34 refer to:

33. Kester, M., Karpa, K. D. & Vrana, K. E. Renal System. in *Elsevier's Integrated Review Pharmacology (Second Edition)* (eds. Kester, M., Karpa, K. D. & Vrana, K. E.) 153–160 (W.B. Saunders, 2012). doi:<https://doi.org/10.1016/B978-0-323-07445-2.00009-4>
34. Stamper, R. L., Lieberman, M. F. & Drake, M. V. Carbonic anhydrase inhibitors. in *Becker-Shaffer's Diagnosis and Therapy of the Glaucomas (Eighth Edition)* (eds. Stamper, R. L., Lieberman, M. F. & Drake, M. V) 407–419 (Mosby, 2009). doi:<https://doi.org/10.1016/B978-0-323-02394-8.00026-7>

- Description of the structure refinement in lines 148-149 seems inconsistent with the methods section. The final refinement step was a reciprocal space refinement, right? That's what is described in the methods.

Response: We did use reciprocal space refinement as described in the methods section, we corrected this accordingly for lines 163-164: “A final structural model was obtained after restrained reciprocal space refinement of the inhibitor-bound protein, fixing several geometry outliers and placing solvent molecules.”

As stated, I believe this paper adds value to the rapidly-growing field of microED by demonstrating a viable use case for the maturing technique, and will also capture the interest of the drug discovery community by describing an attractive pipeline for in-house structure determination to support small-molecule development. I support its publication, provided the authors can address the above concerns. Finally, I commend the authors for posting their manuscript as a preprint simultaneous with peer-review, and I urge them to also check any comments on the preprint server that might improve the quality of the manuscript.

I review non-anonymously,

Michael Thompson
Assistant Professor, UC Merced

Yours sincerely,

Max Clabbers, on behalf of the authors.

Fig. 2 Structure solution and ligand fitting. Electrostatic potential maps displayed for residues of the HCA II active site for the native structure (**a,b**) and the AZM inhibitor bound model (**c,d**). **a.** Initial map after rigid body refinement of the molecular replacement solution for the native model, showing clear difference potential indicative of the Zn²⁺ metal cofactor. **b.** Refined native model after main chain rebuilding and fitting the metal cofactor in the difference map. **c.** Initial map after rigid body refinement of the molecular replacement solution for the ligand-bound model, showing clear difference potential indicative of the Zn²⁺ metal cofactor and AZM inhibitor. **d.** The difference map after rebuilding of the main chain and fitting of the metal was used for placing the drug inhibitor, the found and fitted ligand is superimposed on the structure model, showing its fit to the mFo-DFc map.

Electrostatic potential maps $2mFo-DFc$ are contoured at 1.2σ , colored in blue, and difference maps $mFo-DFc$ contoured at 2.8σ , colored in green and red for positive and negative peaks, respectively. Only observed data were used and no missing reflections were restored for map calculations. Carbon, nitrogen, oxygen and sulphur atoms are colored grey, blue, red and yellow, respectively. Zinc is shown as a dark grey sphere.

Fig. 3 Binding interactions of the AZM inhibitor to the active site of HCA II. **a.** Interatomic distances measured for binding of the sulfonamide group of AZM, the lone pair of the sulfonamide nitrogen is at a distance of 2.1 Å to the active-site zinc, and hydrogen bonding to Thr199 is facilitated at 2.6 Å distance measured from nitrogen to oxygen. **b.** Electrostatic potential map $2mFo-DFc$ contoured at 1.2σ , colored in blue, and difference potential map $mFo-DFc$ contoured at 2.8σ , colored in green and red for positive and negative peaks, respectively. **c.** Simulated annealing (SA) composite omit map calculated with sequentially omitting 5% fractions of the structure, the SA composite omit map is contoured at 1.2σ , colored in magenta. For panels **a-c**, carbon, nitrogen, oxygen and sulphur atoms are colored grey, blue, red and yellow, respectively. Zinc is shown as a dark grey sphere, water shown as a red sphere. Only observed data were used and no missing reflections were restored for map calculations.

Fig. 4 Model and map comparison of HCA II:AZM complex determined by electron, neutron, and x-ray diffraction. **a.** Electrostatic potential map from the MicroED data is shown in blue, contoured at 1.2σ (2.5 Å resolution, 80.0% completeness). **b.** Nuclear density map for joint refinement is shown in yellow, contoured at 1.2σ (PDB ID 4g0c, 2.0 Å resolution, 85.7% completeness). **c.** Electron density map is shown in magenta, contoured at 1.2σ (PDB ID 3hs4, 1.1 Å resolution, 94.7% completeness). **d.** Overlay of the AZM ligand for the MicroED, neutron, and x-ray models. The models were aligned on the zinc coordination sphere including the three active site histidine residues. Carbon atoms are shown in grey, green and cyan for the MicroED, neutron, and x-ray models, respectively. Hydrogen, nitrogen, oxygen and sulphur atoms are colored white, blue, red and yellow, respectively. Zinc is shown as a dark grey sphere.

REVIEWERS' COMMENTS:

Reviewer #3 (Remarks to the Author):

The authors have done an outstanding job of addressing the concerns I raised in my review. I think the final manuscript is excellent, and I highly recommend publication in its current form.

It was a pleasure to review this paper!